# On the Connection between Local Attention and Dynamic Depth-wise Convolution

**Qi Han**[1*]  **Zejia Fan**[2*]  **Qi Dai**[3†]  **Lei Sun**[3]  **Ming-Ming Cheng**[1]  **Jiaying Liu**[2]
**Jingdong Wang**[4†]

TKLNDST, CS, Nankai Univerisy[1], Peking University[2], Microsoft Research Asia[3], Baidu Inc.[4]

## Abstract

Vision Transformer (ViT) attains state-of-the-art performance in visual recognition, and the variant, Local Vision Transformer, makes further improvements. The major component in Local Vision Transformer, local attention, performs the attention separately over small local windows. We rephrase local attention as a channel-wise locally-connected layer and analyze it from two network regularization manners, sparse connectivity and weight sharing, as well as dynamic weight computation.

We point out that local attention resembles depth-wise convolution and its dynamic variants in sparse connectivity: there is no connection across channels, and each position is connected to the positions within a small local window. The main differences lie in (i) weight sharing - depth-wise convolution shares connection weights (kernel weights) across spatial positions and attention shares the connection weights across channels, and (ii) dynamic weight computation manners - local attention is based on dot-products between pairwise positions in the local window, and dynamic convolution is based on linear projections conducted on the center representation or the globally pooled representation.

The connection between local attention and dynamic depth-wise convolution is empirically verified by the ablation study about weight sharing and dynamic weight computation in Local Vision Transformer and (dynamic) depth-wise convolution. We empirically observe that the models based on depth-wise convolution and the dynamic variants with lower computation complexity perform on-par with or slightly better than Swin Transformer, an instance of Local Vision Transformer, for ImageNet classification, COCO object detection and ADE semantic segmentation. Code is available at https://github.com/Atten4Vis/DemystifyLocalViT.

## 1 Introduction

Vision Transformer (Chu et al., 2021b; d'Ascoli et al., 2021; Dosovitskiy et al., 2021; Guo et al., 2021; Han et al., 2020; Khan et al., 2021; Touvron et al., 2020; Wang et al., 2021b; Wu et al., 2021; Xu et al., 2021; Yuan et al., 2021b) has shown promising performance in ImageNet classification. The improved variants, Local Vision Transformer (Chu et al., 2021a; Liu et al., 2021b; Vaswani et al., 2021), adopt the local attention mechanism, which partitions the image space into a set of small windows, and conducts the attention over the windows simultaneously. Local attention leads to great improvement in memory and computation efficiency and makes the extension to downstream tasks easier and more efficient, such as object detection and semantic segmentation.

We exploit the network regularization schemes (Goodfellow et al., 2016), sparse connectivity that controls the model complexity, and weight sharing that relaxes the requirement of increasing the training data scale and reduces the model parameters, as well as dynamic weight prediction that increases the model capability, to study the local attention mechanism. We rephrase local attention as a channel-wise spatially-locally connected layer with dynamic connection weights. The main properties are summarized as follows. (i) Sparse connectivity: there is no connection across channels, and each output position is only connected to the input positions within a local window. (ii) Weight

---

[*]Equal contribution
[†]Corresponding author. wangjingdong@outlook.com

sharing: the connection weights are shared across channels or within each group of channels. (iii) Dynamic weight: the connection weights are dynamically predicted according to each image instance.

We connect local attention with depth-wise convolution (Chollet, 2017; Howard et al., 2017) and its dynamic variants that are also a channel-wise spatially-locally connected layer with optional dynamic connection weights. They are similar in sparse connectivity. The main differences lie in (i) weight sharing - depth-wise convolution shares connection weights (kernel weights) across spatial positions and attention shares the connection weights across channels, and (ii) dynamic weight computation manners - local attention is based on dot-products between pairwise positions in the local window, and dynamic convolution is based on linear projections conducted on the center representation or the globally pooled representation.

We further present the empirical verification for the connection. We take the recently-developed Local Vision Transformer, Swin Transformer (Liu et al., 2021b), as an example, and study the empirical performance of local attention and (dynamic) depth-wise convolution in the same training settings as Swin Transformer. We replace the local attention layer with the (dynamic) depth-wise convolution layer, keeping the overall structure unchanged.

The results show that the (dynamic) depth-wise convolution-based approaches achieve comparable or slightly higher performance for ImageNet classification and two downstream tasks, COCO object detection and ADE semantic segmentation, and (dynamic) depth-wise convolution takes lower computation complexity. The ablation studies imply that weight sharing and dynamic weight improves the model capability. Specifically, (i) for Swin Transformer, weight sharing across channels is beneficial mainly for reducing the parameter (attention weight) complexity, and the attention-based dynamic weight scheme is advantageous in learning instance-specific weights and block-translation equivalent representations; (ii) for depth-wise convolution, weight sharing across positions is beneficial for reducing the parameter complexity as well as learning translation equivalent representations, and the linear projection-based dynamic weight scheme learns instance-specific weights.

## 2 CONNECTING LOCAL ATTENTION AND DEPTH-WISE CONVOLUTION

### 2.1 LOCAL ATTENTION

Vision Transformer (Dosovitskiy et al., 2021) forms a network by repeating the attention layer and the subsequent point-wise MLP (point-wise convolution). The local Vision Transformer, such as Swin Transformer (Liu et al., 2021b) and HaloNet (Vaswani et al., 2021), adopts the local attention layer, which partitions the space into a set of small windows and performs the attention operation within each window simultaneously, to improve the memory and computation efficiency.

The local attention mechanism forms the keys and values in a window that the query lies in. The attention output for the query $\mathbf{x}_i \in \mathbb{R}^D$ at the position $i$ is the aggregation of the corresponding values in the local window, $\{\mathbf{x}_{i1}, \mathbf{x}_{i2}, \ldots, \mathbf{x}_{iN_k}\}$, weighted by the corresponding attention weights $\{a_{i1}, a_{i2}, \ldots, a_{iN_k}\}$[1]:

$$\mathbf{y}_i = \sum_{j=1}^{N_k} a_{ij}\mathbf{x}_{ij}, \tag{1}$$

where $N_k = K_w \times K_h$ is the size of the local window. The attention weight $a_{ij}$ is computed as the softmax normalization of the dot-product between the query $\mathbf{x}_i$ and the key $\mathbf{x}_{ij}$:

$$a_{ij} = \frac{e^{\frac{1}{\sqrt{D}}\mathbf{x}_i^\top \mathbf{x}_{ij}}}{Z_i} \quad \text{where} \quad Z_i = \sum_{j=1}^{N_k} e^{\frac{1}{\sqrt{D}}\mathbf{x}_i^\top \mathbf{x}_{ij}}. \tag{2}$$

The multi-head version partitions the $D$-dimensional query, key and value vectors into $M$ subvectors (each with $\frac{D}{M}$ dimensions), and conducts the attention process $M$ times, each over the corresponding subvector. The whole output is the concatenation of $M$ outputs, $\mathbf{y}_i = [\mathbf{y}_{i1}^\top \ \mathbf{y}_{i2}^\top \ \cdots \ \mathbf{y}_{iM}^\top]^\top$. The $m$th output $\mathbf{y}_{im}$ is calculated by

$$\mathbf{y}_{im} = \sum_{j=1}^{N_k} a_{ijm}\mathbf{x}_{ijm}, \tag{3}$$

where $\mathbf{x}_{ijm}$ is the $m$th value subvector and $a_{ijm}$ is the attention weight computed from the $m$th head in the same way as Equation 2.

---

[1]For presentation convenience, we ignore the linear projections conducted to the queries, the keys and the values. In vision applications, the value and the corresponding key are from the same feature possibly with different linear projections, and we denote them using the same symbol $\mathbf{x}_{ij}$.

Figure 1: Illustration of connectivity for (a) convolution, (b) global attention and spatial mixing MLP, (c) local attention and depth-wise convolution, (d) point-wise MLP or $1 \times 1$ convolution, and (e) MLP (fully-connected layer). In the spatial dimension, we use 1D to illustrate the local-connectivity pattern for clarity.

## 2.2 SPARSE CONNECTIVITY, WEIGHT SHARING, AND DYNAMIC WEIGHT

We give a brief introduction of two regularization forms, sparse connectivity and weight sharing, and dynamic weight, and their benefits. We will use the three forms to analyze local attention and connect it to dynamic depth-wise convolution.

*Sparse connectivity* means that there are no connections between some output neurons (variables) and some input neurons in a layer. It reduces the model complexity without decreasing the number of neurons, e.g., the size of the (hidden) representations.

*Weight sharing* indicates that some connection weights are equal. It lowers the number of model parameters and increases the network size without requiring a corresponding increase in training data (Goodfellow et al., 2016).

*Dynamic weight* refers to learning specialized connection weights for each instance. It generally aims to increase the model capacity. If regarding the learned connection weights as hidden variables, dynamic weight can be viewed as introducing second-order operations that increase the capability of the network. The connection to Hopfield networks is discussed in (Ramsauer et al., 2020).

## 2.3 ANALYZING LOCAL ATTENTION

We show that local attention is a channel-wise spatially-locally connected layer with dynamic weight computation, and discuss its properties. Figure 1 (c) illustrates the connectivity pattern.

The aggregation processes (Equation 1 and Equation 3) for local attention can be rewritten equivalently in a form of element-wise multiplication:

$$\mathbf{y}_i = \sum_{j=1}^{N_k} \mathbf{w}_{ij} \odot \mathbf{x}_{ij}, \tag{4}$$

where $\odot$ is the element-wise multiplication operator, and $\mathbf{w}_{ij} \in \mathbb{R}^D$ is the weight vector formed from the attention weight $a_{ij}$ or $\{a_{ij1}, a_{ij2}, \ldots, a_{ijM}\}$.

*Sparse connectivity.* The local attention layer is spatially sparse: each position is connected to the $N_k$ positions in a small local window. There are also no connections across channels. The element-wise multiplication in Equation 4 indicates that given the attention weights, each output element, e.g., $y_{id}$ (the $i$th position for the $d$th channel), is only dependent on the corresponding input elements from the same channel in the window, $\{x_{i1d}, x_{i2d}, \ldots, x_{iN_kd}\}$, and not related to other channels.

*Weight sharing.* The weights are shared with respect to channels. In the single-head attention case, all the elements $\{w_{ij1}, w_{ij2}, \ldots, w_{ijD}\}$ in the weight vector $\mathbf{w}_{ij}$ are the same: $w_{ijd} = a_{ij}, 1 \leqslant d \leqslant D$. In the multi-head attention case, the weight vector $\mathbf{w}_{ij}$ is group-wise same: $\mathbf{w}_{ij}$ is partitioned to $M$ subvectors each corresponding to one attention head, $\{\mathbf{w}_{ij1}, \mathbf{w}_{ij2}, \ldots, \mathbf{w}_{ijM}\}$, and the elements in each subvector $\mathbf{w}_{ijm}$ are the same and are equal to the $m$th attention weight, $a_{ijm}$.

*Dynamic weight.* The weights, $\{\mathbf{w}_{i1}, \mathbf{w}_{i2}, \ldots, \mathbf{w}_{iN_k}\}$, are dynamically predicted from the query $\mathbf{x}_i$ and the keys $\{\mathbf{x}_{i1}, \mathbf{x}_{i2}, \ldots, \mathbf{x}_{iN_k}\}$ in the local window as shown in Equation 2. We rewrite it as:

$$\{\mathbf{w}_{i1}, \mathbf{w}_{i2}, \ldots, \mathbf{w}_{iN_k}\} = f(\mathbf{x}_i; \mathbf{x}_{i1}, \mathbf{x}_{i2}, \ldots, \mathbf{x}_{iN_k}). \tag{5}$$

Each weight may obtain the information across all the channels in one head, and serves as a bridge to deliver the across-channel information to each output channel.

*Translation equivalence.* Different from convolution which satisfies translation equivalence through sharing weights across positions, the equivalence to translation for local attention, depends if the keys/values are changed, i.e., the attention weights are changed, when the feature map is translated.

In the case of sparsely-sampled window (for run-time efficiency), e.g., (Hu et al., 2019; Liu et al., 2021b; Ramachandran et al., 2019; Vaswani et al., 2021), local attention is equivalent to block-wise translation, i.e., the translation is a block or multiple blocks with the block size same as the window size $K_w \times K_h$, and otherwise not equivalent (as keys/values are changed). In the case that the windows are densely sampled (e.g., (Zhao et al., 2020)), local attention is equivalent to translation.

*Set representation.* The keys/values for one query are collected as a set with the spatial-order information lost. This leads to that the spatial correspondence between the keys/values across windows is not exploited. The order information loss is partially remedied by encoding the positions as embeddings (Dosovitskiy et al., 2021; Touvron et al., 2020), or learning a so-called relative position embedding (e.g., (Liu et al., 2021b)) in which the spatial-order information is preserved as the keys/values in a local window are collected as a vector.

## 2.4 CONNECTION TO DYNAMIC DEPTH-WISE CONVOLUTION

Depth-wise convolution is a type of convolution that applies a single convolutional filter for each channel: $\bar{\mathbf{X}}_d = \mathbf{C}_d \otimes \mathbf{X}_d$, where $\mathbf{X}_d$ and $\bar{\mathbf{X}}_d$ are the $d$th input and output channel maps, $\mathbf{C}_d \in \mathbb{R}^{N_k}$ is the corresponding kernel weight, and $\otimes$ is the convolution operation. It can be equivalently written in the form of element-wise multiplication for each position:

$$\mathbf{y}_i = \sum_{j=1}^{N_k} \mathbf{w}_{\text{offset}(i,j)} \odot \mathbf{x}_{ij}. \tag{6}$$

Here, $\text{offset}(i, j)$ is the relative offset, from the 2D coordinate of the position $j$ to the 2D coordinate of the central position $i$. The weights $\{\mathbf{w}_{\text{offset}(i,j)} \in \mathbb{R}^D; j = 1, 2, \ldots, N_k\}$ are reshaped from $\mathbf{C}_1, \mathbf{C}_2, \ldots, \mathbf{C}_D$. The $N_k$ weight vectors are model parameters and shared for all the positions.

We also consider two dynamic variants of depth-wise convolution: homogeneous and inhomogeneous[2]. The homogeneous dynamic variant predicts the convolution weights using linear projections from a feature vector that is obtained by globally-pooling the feature maps:

$$\{\mathbf{w}_1, \mathbf{w}_2, \ldots, \mathbf{w}_{N_k}\} = g(\text{GAP}(\mathbf{x}_1, \mathbf{x}_2, \ldots, \mathbf{x}_N)). \tag{7}$$

Here, $\{\mathbf{x}_1, \mathbf{x}_2, \ldots, \mathbf{x}_N\}$ are the image responses. $\text{GAP}()$ is the global average pooling operator. $g()$ is a function based on linear projection: a linear projection layer to reduce the channel dimension with BN and ReLU, followed by another linear projection to generate the connection weights.

The inhomogeneous dynamic variant predicts the convolution weights separately for each position from the feature vector $\mathbf{x}_i$ at the position (the center of the window):

$$\{\mathbf{w}_{i_1}, \mathbf{w}_{i_2}, \ldots, \mathbf{w}_{i_{N_k}}\} = g(\mathbf{x}_i). \tag{8}$$

This means that the weights are not shared across positions. We share the weights across the channels in a way similar to the multi-head attention mechanism to reduce the complexity.

We describe the similarities and differences between (dynamic) depth-wise convolution and local attention. Figure 1 (c) illustrates the connectivity patterns and Table 1 shows the properties between local attention and depth-wise convolution , and various other modules.

*Similarity.* Depth-wise convolution resembles local attention in *sparse connectivity*. There are no connections across channels. Each position is only connected to the positions in a small local window for each channel.

*Difference.* One main difference lies in *weight sharing*: depth-wise convolution shares the connection weights across spatial positions, while local attention shares the weights across channels or within each group of channels. Local attention uses proper weight sharing across channels to get better performance. Depth-wise convolution benefits from the weight sharing across positions to reduce the parameter complexity and increase the network capability.

The second difference is that the connection weights for depth-wise convolution are *static* and learned as model parameters, while the connection weights for local attention are *dynamic* and predicted from each instance. The *dynamic* variants of depth-wise convolution also benefit from the dynamic weight.

---

[2]The homogeneous version follows and applies dynamic convolution to depth-wise convolution. The inhomogeneous version is close to involution (Li et al., 2021) and lightweight depth-wise convolution (Wu et al., 2019).

Table 1: The comparison of attention, local MLP (non-dynamic version of local attention, the attention weights are learned as static model parameters), local attention, convolution, depth-wise convolution (DW-Conv.) and the dynamic variant (D-DW-Conv.) in terms of the patterns of sparse connectivity, weight sharing, and dynamic weight. Please refer to Figure 1 for the connectivity pattern illustration.

| | Sparse between positions | | Sparse between | Weight sharing across | | Dynamic |
| | non-local | full | channels | position | channel | weight |
|---|---|---|---|---|---|---|
| Local MLP | ✓ | | ✓ | | ✓[♭] | |
| Local attention | ✓ | | ✓ | | ✓[♭] | ✓ |
| DW-Conv. | ✓ | | ✓ | ✓ | | |
| D-DW-Conv. | ✓ | | ✓ | ✓ | | ✓ |
| Conv. | ✓ | | | ✓ | | |

One more difference lies in window representation. Local attention represents the positions in a window by utilizing a *set* form with spatial-order information lost. It explores the spatial-order information implicitly using the positional embedding or explicitly using the learned so-called relative positional embedding. Depth-wise convolution exploits a *vector* form: aggregate the representations within a local window with the weights indexed by the relative position (see Equation 6); keep spatial correspondence between the positions for different windows, thus exploring the spatial-order information explicitly.

## 3   EXPERIMENTAL STUDY

We conduct empirical comparisons between local attention and depth-wise convolutions on three visual recognition tasks: ImageNet classification, COCO object detection, and ADE semantic segmentation. We follow the structure of Swin Transformer to build the depth-wise convolution-based networks. We apply the same training and evaluation settings from Swin Transformer to our models. In addition, we study the effects of weight sharing and dynamic weight in the two structures. The results for large scale pre-training are given in the appendix.

### 3.1   ARCHITECTURES

We use the recently-developed Swin Transformer as the example of local attention-based networks and study the performance over the tiny and base networks: Swin-T and Swin-B, provided by the authors (Liu et al., 2021b) We follow the tiny and base networks to build two depth-wise convolution-based networks, DW-Conv.-T and DW-Conv.-B so that the overall architectures are the same, making the comparison fair. We also build two dynamic versions, D-DW-Conv. and I-D-DW-Conv., by predicting the dynamic weights as described in Section 2.4. We simply replace local attention in Swin Transformer by depth-wise convolution of the same window size, where the pre- and post-linear projections over the values are replaced by $1 \times 1$ convolutions. We adopt the convolutional network design pattern to append BN (Ioffe & Szegedy, 2015) and ReLU (Nair & Hinton, 2010) to the convolution. The details are available in the Appendix. In terms of parameter and computation complexity, the depth-wise convolution-based networks are lower (Table 2) because there are linear projections for keys and values in local attention.

### 3.2   DATASETS AND IMPLEMENTATION DETAILS

**ImageNet classification.** The ImageNet-1K recognition dataset (Deng et al., 2009) contains 1.28M training images and 50K validation images with totally 1,000 classes. We use the exactly-same training setting as Swin Transformer (Liu et al., 2021b). The AdamW (Loshchilov & Hutter, 2019) optimizer for 300 epochs is adopted, with a cosine decay learning rate scheduler and 20 epochs of linear warm-up. The weight decay is 0.05, and the initial learning rate is 0.001. The augmentation and regularization strategies include RandAugment (Cubuk et al., 2020), Mixup (Zhang et al., 2018a), CutMix (Yun et al., 2019), stochastic depth (Huang et al., 2016), etc.

**COCO object detection.** The COCO 2017 dataset (Lin et al., 2014) contains 118K training and 5K validation images. We follow Swin Transformer to adopt Cascade Mask R-CNN (Cai & Vasconcelos, 2019) for comparing backbones. We use the training and test settings from Swin Transformer: multi-scale training - resizing the input such that the shorter side is between 480 and 800 and the longer side is at most 1333; AdamW optimizer with the initial learning rate 0.0001; weight decay - 0.05; batch size - 16; and epochs - 36.

**ADE semantic segmentation.** The ADE20K (Zhou et al., 2017) dataset contains 25K images, 20K for training, 2K for validation, and 3K for testing, with 150 semantic categories. The same setting

Table 2: ImageNet classification comparison for ResNet, Mixer and ResMLP, ViT and DeiT, Swin (Swin Transformer), DW-Conv. (depth-wise convolution), and D-DW-Conv. (dynamic depth-wise convolution).

| method | img. size | #param. | FLOPs | throughput (img. / s) | top-1 acc. | real acc. |
|---|---|---|---|---|---|---|
| *Bottleneck: convolution with low rank* | | | | | | |
| ResNet-50 (He et al., 2016) | $224^2$ | 26M | 4.1G | 1128.3 | 76.2 | 82.5 |
| ResNet-101 (He et al., 2016) | $224^2$ | 45M | 7.9G | 652.0 | 77.4 | 83.7 |
| ResNet-152 (He et al., 2016) | $224^2$ | 60M | 11.6G | 456.7 | 78.3 | 84.1 |
| *Channel and spatial separable MLP, spatial separable MLP = point-wise $1 \times 1$ convolution* | | | | | | |
| Mixer-B/16 (Tolstikhin et al., 2021) | $224^2$ | 46M | - | - | 76.4 | 82.4 |
| Mixer-L/16 (Tolstikhin et al., 2021) | $224^2$ | 189M | - | - | 71.8 | 77.1 |
| ResMLP-12 (Touvron et al., 2021) | $224^2$ | 15M | 3.0G | - | 76.6 | 83.3 |
| ResMLP-24 (Touvron et al., 2021) | $224^2$ | 30M | 6.0G | - | 79.4 | 85.3 |
| ResMLP-36 (Touvron et al., 2021) | $224^2$ | 45M | 8.9G | - | 79.7 | 85.6 |
| *Global attention: dynamic channel separable MLP + spatial separable MLP* | | | | | | |
| ViT-B/16 (Dosovitskiy et al., 2021) | $384^2$ | 86M | 55.4G | 83.4 | 77.9 | 83.6 |
| ViT-L/16 (Dosovitskiy et al., 2021) | $384^2$ | 307M | 190.7G | 26.5 | 76.5 | 82.2 |
| DeiT-S (Touvron et al., 2020) | $224^2$ | 22M | 4.6G | 947.3 | 79.8 | 85.7 |
| DeiT-B (Touvron et al., 2020) | $224^2$ | 86M | 17.5G | 298.2 | 81.8 | 86.7 |
| DeiT-B (Touvron et al., 2020) | $384^2$ | 86M | 55.4G | 82.7 | 83.1 | 87.7 |
| *Local MLP: perform static separable MLP in local small windows* | | | | | | |
| Swin-Local MLP-T | $224^2$ | 26M | 3.8G | 861.0 | 80.3 | 86.1 |
| Swin-Local MLP-B | $224^2$ | 79M | 12.9G | 321.2 | 82.2 | 86.9 |
| *Local attention: perform attention in local small windows* | | | | | | |
| Swin-T (Liu et al., 2021b) | $224^2$ | 28M | 4.5G | 713.5 | 81.3 | 86.6 |
| Swin-B (Liu et al., 2021b) | $224^2$ | 88M | 15.4G | 263.0 | 83.3 | 87.9 |
| *Depth-wise convolution + point-wise $1 \times 1$ convolution* | | | | | | |
| DW-Conv.-T | $224^2$ | 24M | 3.8G | 928.7 | 81.3 | 86.8 |
| DW-Conv.-B | $224^2$ | 74M | 12.9G | 327.6 | 83.2 | 87.9 |
| D-DW-Conv.-T | $224^2$ | 51M | 3.8G | 897.0 | 81.9 | 87.3 |
| D-DW-Conv.-B | $224^2$ | 162M | 13.0G | 322.4 | 83.2 | 87.9 |
| I-D-DW-Conv.-T | $224^2$ | 26M | 4.4G | 685.3 | 81.8 | 87.1 |
| I-D-DW-Conv.-B | $224^2$ | 80M | 14.3G | 244.9 | 83.4 | 88.0 |

as Swin Transformer (Liu et al., 2021b) is adopted. UPerNet (Xiao et al., 2018) is used as the segmentation framework. Details are provided in the Appendix.

## 3.3 RESULTS

**ImageNet classification.** The comparison for ImageNet classification is given in Table 2. One can see that the local attention-based networks, Swin Transformer, and the depth-wise convolution-based networks, perform on par (with a slight difference of 0.1) in terms of top-1 accuracy and real accuracy (Beyer et al., 2020) for both tiny and base models. In the tiny model case, the two dynamic depth-wise convolution-based networks perform higher. In particular, the depth-wise convolution-based networks are more efficient in parameters and computation complexities. In the tiny model case, the parameters and computation complexities are reduced by 14.2% and 15.5%, respectively. Similarly, in the base model case, the two costs are reduced by 15.9% and 16.2%, respectively. The homogeneous dynamic variant takes more parameters but with almost the same complexity efficiency, and the inhomogeneous dynamic variant take advantage of weight sharing across channels that reduce the model parameters.

**COCO object detection.** The comparisons between local attention (Swin Transformer), depth-wise convolution, and two versions of dynamic depth-wise convolution are shown in Table 3. Depth-wise convolution performs a little lower than local attention, and dynamic depth-wise convolution performs better than the static version and on par with local attention.

**ADE semantic Segmentation.** The comparisons of single scale testing on ADE semantic segmentation are shown in Table 3. In the tiny model case, (dynamic) depth-wise convolution is ~1.0% higher than local attention. In the base model case, the performances are similar[3].

---

[3]We conducted an additional experiment by changing the ending learning rate from 0 to $1e - 6$. The base model with depth-wise convolutions achieves a higher mIoU score: 48.9.

Table 3: Comparison results on COCO object detection and ADE semantic segmentation.

| | COCO Object Detection | | | | | | ADE20K Semantic Segmentation | | |
|---|---|---|---|---|---|---|---|---|---|
| | #param. | FLOPs | $AP^{box}$ | $AP^{box}_{50}$ | $AP^{box}_{75}$ | $AP^{mask}$ | #param. | FLOPs | mIoU |
| Swin-T | 86M | 747G | 50.5 | 69.3 | 54.9 | 43.7 | 60M | 947G | 44.5 |
| DW Conv.-T | 82M | 730G | 49.9 | 68.6 | 54.3 | 43.4 | 56M | 928G | 45.5 |
| D-DW Conv.-T | 108M | 730G | 50.5 | 69.5 | 54.6 | 43.7 | 83M | 928G | 45.7 |
| I-D-DW Conv.-T | 84M | 741G | 50.8 | 69.5 | 55.3 | 44.0 | 58M | 939G | 46.2 |
| Swin-B | 145M | 986G | 51.9 | 70.9 | 56.5 | 45.0 | 121M | 1192G | 48.1 |
| DW Conv.-B | 132M | 924G | 51.1 | 69.6 | 55.4 | 44.2 | 108M | 1129G | 48.3 |
| D-DW Conv.-B | 219M | 924G | 51.2 | 70.0 | 55.4 | 44.4 | 195M | 1129G | 48.0 |
| I-D-DW Conv.-B | 137M | 948G | 51.8 | 70.3 | 56.1 | 44.8 | 114M | 1153G | 47.8 |

Table 4: Effects of weight sharing across channels and positions. The results are reported on the ImageNet top-1 accuracy. SC = Sharing across channels. SP = sharing across positions.

| | SC | SP | Acc. | #param. | | SC | SP | Acc. | #param. |
|---|---|---|---|---|---|---|---|---|---|
| | ✗ | ✓ | 80.2 | 35.3M | | ✗ | ✓ | 81.3 | 24.2M |
| Local MLP | ✓ | ✗ | 80.3 | 26.2M | DW Conv. | ✓ | ✗ | 80.3 | 26.2M |
| | ✓ | ✓ | 80.3 | 24.3M | | ✓ | ✓ | 81.1 | 23.9M |

## 3.4 EMPIRICAL ANALYSIS

Local and channel-separable connection has been shown to be helpful for visual recognition. The empirical results in Table 2, e.g., local attention performs better than global attention (local connection) and depth-wise convolution performs better than convolution (channel-separable connection), also verify it. In the following, we present empirical results for weight sharing and dynamic weight by taking the tiny models as examples.

**Weight sharing.** We study how the performance is affected by the number of channels in each group across which the weights are shared (the numbers of attention heads at each stage are accordingly changed) for local attention and local MLP (learn the weights in each window as model parameters and not shared across windows). Figure 2 shows the effect for (a) local MLP - static weights, and (b) local attention - dynamic weights. One can see that for local attention, too many channels and too few channels in each group perform similarly, but do not lead to the best. For local MLP, weight sharing significantly reduces model parameters. These indicate proper weight sharing across channels is helpful for both local attention and local MLP.

We further study the effect of combining the weight sharing pattern for local MLP and depth-wise convolution. For local MLP, Weight sharing across positions means the connection weight is shared for different spatial blocks in local MLP. For convolution, the scheme of sharing weights across channels is similar to the multi-head manner in local attention. The results in Table 4 suggest that: (i) for local MLP, sharing weight across channels reduces the model parameters and sharing across spatial blocks do not have big impact; (ii) For depth-wise convolution, sharing weight across channels does not have big impact, but sharing weight across positions significantly increase the performance.

The window sampling scheme for local MLP and depth-wise convolution is different: local MLP sparsely samples the windows using the way in Swin Transformer, for reducing the high memory cost, and depth-wise convolution densely sample the windows. Weight sharing across positions in local MLP is insufficient for learning translation-equivalent representation, explaining why local MLP with weight sharing across both channels and positions performs lower than depth-wise convolution with additional weight sharing across channels.

**Dynamic weight.** We study how dynamic weight in local attention affects performance. As seen from Table 2, local MLP achieves, the static version, $80.3\%$ and $82.2\%$ for tiny and base models, lower than Swin, the dynamic version, $81.3\%$ and $83.3\%$. This implies that dynamic weight is helpful. The improvements from dynamic weight are also observed for depth-wise convolution (Table 2).

We further study the effects of the attention scheme and the linear-projection scheme for dynamic weight computation. The observations from in Table 5 include: the attention mechanism for shifted and sliding window sampling performs similarly; the inhomogeneous dynamic weight computation way is better than the attention mechanism ($81.8$ vs $81.4$). We think that the reasons for the latter observation include: for the attention mechanism the representation is only block translation equivalent other than any translation equivalent; the linear projection-based dynamic weight scheme

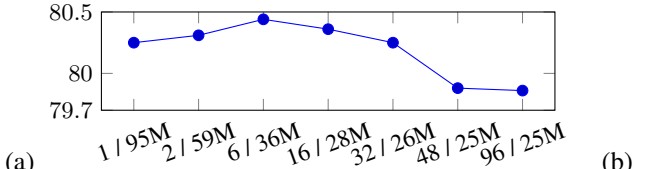 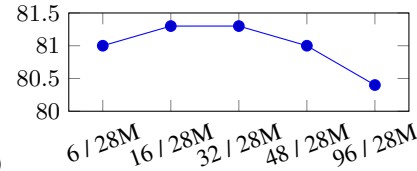

(a)
(b)

Figure 2: Effect of #channels sharing the weights on ImageNet classification. X-axis: #channels within each group / #param. Y-axis: ImageNet classification accuracy. (a) Local MLP: the static version of Swin transformer. (b) Local attention: Swin transformer. Results is reported for tiny model on ImageNet dataset.

Table 5: Comparison of different dynamic weight manners. The results are reported on the ImageNet top-1 accuracy. Shifted window sampling (Win. samp.) means the way in Swin Transformer and sliding means the densely-sampling manner. The result of Sliding local MLP is from (Liu et al., 2021b). homo. dyna. = homogeneous dynamic weight. inhomo. dyna. = inhomogeneous dynamic weight.

| | Win. samp. | #param. | FLOPs | Acc. | | Win. samp. | #param. | FLOPs | Acc. |
|---|---|---|---|---|---|---|---|---|---|
| Local MLP | shifted | 26M | 3.8G | 80.3 | DW Conv. | sliding | 24M | 3.8G | 81.3 |
| w/ attention | shifted | 28M | 4.5G | 81.3 | w/ homo. dyna. | sliding | 51M | 3.8G | 81.9 |
| w/ attention | sliding | 28M | 4.5G | 81.4 | w/ inhomo. dyna. | sliding | 26M | 4.4G | 81.8 |

(vector representation for the window) learns better weights than the attention-based scheme (set representation for the window). We also observe that such influence is eliminated for large models and detection tasks.

**Set representation.** Local attention represents the positions in a window as a set with the spatial-order information lost. Swin Transformer learns relative positional embeddings where the positions in a window are actually described as a vector keeping the spatial-order information. It is reported in (Liu et al., 2021b) that removing the relative positional embeddings leads to a $1.2\%$ accuracy drop, indicating the spatial-order information is important.

**Concurrent works.** We give the comparison between inhomogeneous dynamic depth-wise convolution (I-D-DW Conv.) and concurrent local attention-based works (Chu et al., 2021a; Wang et al., 2021a; Huang et al., 2021; Xu et al., 2021) in Table 6. We follow Shuffle Transformer and add an extra DW Conv. before FFN in I-D-DW Conv, and the performance is improved by $0.5$. The performance is on par with these concurrent works except the Twins-SVT (81.9%, 2.9G) which uses interleaved attention and additional depth-wise convolutions.

## 4 RELATED WORK

**Sparse connectivity.** Sparse connection across channels is widely explored for removing redundancy in the channel domain. The typical schemes are depth-wise convolution adopted by MobileNet (Howard et al., 2017; Sandler et al., 2018), ShuffleNetV2 (Ma et al., 2018) and IGCv3 (Sun et al., 2018), and group convolution adopted by ResNeXt (Xie et al., 2017), merge-and-run (Zhao et al., 2018), ShuffleNetV1 (Zhang et al., 2018b), and IGC (Zhang et al., 2017).

The self-attention unit[4] in Vision Transformer, its variants (Chen et al., 2020; Chu et al., 2021b; Dosovitskiy et al., 2021; Han et al., 2021; Heo et al., 2021; Li et al., 2021; Liu et al., 2021b; Pan et al., 2021; Touvron et al., 2020; Vaswani et al., 2021; Wang et al., 2021b; Wu et al., 2021; Yuan et al., 2021a;b; Zhang et al., 2021; Zhao et al., 2020; Zhou et al., 2021), and the spatial information fusion unit (e.g., token-mixer in MLP-Mixer (Tolstikhin et al., 2021) and ResMLP (Touvron et al., 2021)) have no connections across channels.

$1 \times 1$ (point-wise) convolution (in ShuffleNetV2 (Ma et al., 2018), MobileNet (Howard et al., 2017; Sandler et al., 2018), IGC (Zhang et al., 2017), ViT (Dosovitskiy et al., 2021), local ViT (Liu et al., 2021b; Vaswani et al., 2021), MLP-Mixer (Tolstikhin et al., 2021), ResMLP (Touvron et al., 2021)) has no connections across spatial positions. The convolutions with other kernel sizes and local attention (Zhao et al., 2020; Liu et al., 2021b; Vaswani et al., 2021) have connections between each position and the positions within a small local window, respectively.

---

[4]The pre- and post- linear projections for values can be regarded as $1 \times 1$ convolutions. The attention weights generated from keys and values with linear projections in some sense mix the information across channels.

Table 6: Comparison with concurrent works on ImageNet classification with tiny models.

|  | #param. | FLOPs | top-1 acc. |
|---|---|---|---|
| Twins-PCPVT (Chu et al., 2021a) | 24M | 3.8G | 81.2 |
| Twins-SVT (Chu et al., 2021a) | 24M | 2.9G | 81.7 |
| CoaT-Lite (Xu et al., 2021) | 20M | 4.0G | 81.9 |
| CoaT (Xu et al., 2021) | 22M | 12.6G | 82.1 |
| PVT-v2 (Wang et al., 2021a) | 25M | 4.0G | 82.0 |
| Shuffle Transformer (Huang et al., 2021) | 29M | 4.6G | 82.5 |
| I-D-DW Conv. | 26M | 4.4G | 81.8 |
| I-D-DW Conv. + DW | 27M | 4.4G | 82.3 |

**Weight sharing.** Weight sharing across spatial positions is mainly used in convolution, including normal convolution, depth-wise convolution and point-wise convolution. Weight sharing across channels is adopted in the attention unit (Vaswani et al., 2017), its variants (Chu et al., 2021a;b; Dosovitskiy et al., 2021; Li et al., 2021; Liu et al., 2021b; Touvron et al., 2020; Vaswani et al., 2021; Wang et al., 2021b; Wu et al., 2021; Yuan et al., 2021b), and token-mixer MLP in MLP-mixer (Tolstikhin et al., 2021) and ResMLP (Touvron et al., 2021).

**Dynamic weight.** Predicting the connection weights is widely studied in convolutional networks. There are basically two types. One is to learn homogeneous connection weights, e.g., SENet (Hu et al., 2018b), dynamic convolution (Jia et al., 2016). The other is to learn the weights for each region or each position (GENet (Hu et al., 2018a), Lite-HRNet (Yu et al., 2021), Involution (Li et al., 2021)). The attention unit in ViT or local ViT learns dynamic connection weights for each position.

**Networks built with depth-wise separable convolutions.** There are many networks built upon depth-wise separable convolution or its variants, such as MobileNet (Howard et al., 2017; Sandler et al., 2018), ShuffleNet (Ma et al., 2018), IGC (Zhang et al., 2017), Xception (Chollet, 2017), and EfficientNet (Tan & Le, 2019; 2021). In this paper, our goal is to connect dynamic depth-wise convolution with local attention.

**Convolution vs Transformer.** The study in (Cordonnier et al., 2020) shows that a multi-head self-attention layer can simulate a convolutional layer by developing additional carefully-designed relative positional embeddings with *the attention part dropped*. Differently, we connect (dynamic) depth-wise convolution and local self-attention by *connecting the attention weights for self-attention and the dynamic weights for convolution* (as well as studying weight sharing). In (Andreoli, 2019), the mathematical connection (in terms of the tensor form) between convolution and attention is presented. The opinion that convolution and attention are essentially about the model complexity control is similar to ours, and we make the detailed analysis and report empirical studies.

The concurrently-developed work in NLP (Tay et al., 2021) empirically compares lightweight depth-wise convolution (Wu et al., 2019) to Transformer for NLP tasks, and reaches a conclusion similar to ours for vision tasks: convolution and Transformer obtain on-par results. Differently, we attempt to understand why they perform on par from three perspectives: sparse connectivity, weight sharing and dynamic weight, and discuss their similarities and differences.

## 5 CONCLUSION

The connections between local attention and dynamic depth-wise convolution are summarized as follows. (i) Same with dynamic depth-wise convolution, local attention benefits from two sparse connectivity forms: local connection and no connection across channels. (ii) Weight sharing across channels in local attention is helpful for reducing the parameter (attention weight) complexity and slightly boosting the performance, and weight sharing across positions in depth-wise convolution is helpful for reducing the parameter complexity and learning translation-equivalent representations and thus boosting the performance. (iii) The attention-based dynamic weight computation for local attention is beneficial for learning image-dependent weights and block-translation equivalent representations, and the linear projection-based dynamic weight computation for (in)homogeneous dynamic depth-wise convolution is beneficial for learning image-dependent weights. Inhomogeneous dynamic depth-wise convolution is superior over local attention for ImageNet classification and segmentation in the case of tiny models, and on par for larger models and detection tasks. In addition, the better downstream performance for local attention and depth-wise convolution stems from the larger kernel size ($7 \times 7$ vs $3 \times 3$), which is also observed in Yuan et al. (2021d).

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

# APPENDIX

## A RELATION GRAPH

We present a relation graph in Figure 3 to describe the relation between convolution, depth-wise separable convolution (depth-wise convolution + $1 \times 1$ convolution), Vision Transformer, Local Vision Transformer, as well as multilayer perceptron (MLP), Separable MLP in terms of sparse connectivity, weight sharing, and dynamic weight. Table 7

Multilayer perceptron (MLP) is a fully-connected layer: each neuron (an element at each position and each channel) in one layer is connected with all the neurons in the previous layer[5]. Convolution and separable MLP are sparse versions of MLP. The connection weights can be formulated as a tensor (e.g., 3D tensor, two dimension for space and one dimension for channel) and the low-rank approximation of the tensor can be used to regularize the MLP.

Convolution is a locally-connected layer, formed by connecting each neuron to the neurons in a small local window with the weights shared across the spatial positions. Depth-wise separable convolution is formed by decomposing the convolution into two components: one is point-wise $1 \times 1$ convolution, mixing the information across channels, and the other is depth-wise convolution, mixing the spatial information. Other variants of convolution, such as bottleneck, multi-scale convolution or pyramid, can be regarded as low-rank variants.

Separable MLP (e.g., MLP-Mixer and ResMLP) reshapes the 3D tensor into a 2D format with the spatial dimension and channel dimension. Separable MLP consists of two sparse MLP along the two dimensions separately, which are formed by separating the input neurons into groups. Regarding channel sparsity, the neurons in the same channel form a group, and an MLP is performed over each group with the MLP parameters shared across groups, forming the first sparse MLP (spatial/token mixing). A similar process is done by viewing the neurons at the same position into a group, forming the second sparse MLP (channel mixing).

Vision Transformer is a dynamic version of separable MLP. The weights in the first sparse MLP (spatial/token mixing) are dynamically predicted from each instance. Local Vision Transformer is a spatially-sparser version of Vision Transformer: each output neuron is connected to the input neurons in a local window. PVT (Wang et al., 2021b) is a pyramid (spatial sampling/ low-rank) variant of Vision Transformer.

Depth-wise separable convolution can also be regarded as a spatially-sparser version of separable MLP. In the first sparse MLP (spatial/token mixing), each output neuron is only dependent on the input neurons in a local window, forming depth-wise convolution. In addition, the connection weights are shared across spatial positions, instead of across channels.

## B MATRIX FORM EXPLANATION

We use the matrix form to explain sparsity connectivity in various layers and how they are obtained by modifying the MLP.

**MLP.** The term MLP, Multilayer Perceptron, is used ambiguously, sometimes loosely to any feedforward neural network. We adopt one of the common definitions, and use it to refer to fully-connected layers. Our discussion is based on a single fully-connected layer, and can be easily generalized to two or more fully-connected layers. One major component, except the nonlinear units and others, is a linear transformation:

$$\mathbf{y} = \mathbf{W}\mathbf{x}, \tag{9}$$

where $\mathbf{x}$ represents the input neurons, $\mathbf{y}$ represents the output neurons, and $\mathbf{W}$ represents the connection weights, e.g., $\mathbf{W} \in \mathbb{R}^{NC \times NC}$, where $N$ is the number of positions, and $C$ is the number of channels.

**Convolution.** Considering the 1D case with a single channel (the 2D case is similar), the connection weight matrix $\mathbf{W} \in \mathbb{R}^{N \times N}$ is in the following sparse form, also known as the Toeplitz matrix (We

---

[5]We use the widely-used definition for the term MLP: fully-connected layer. There might be other definitions.

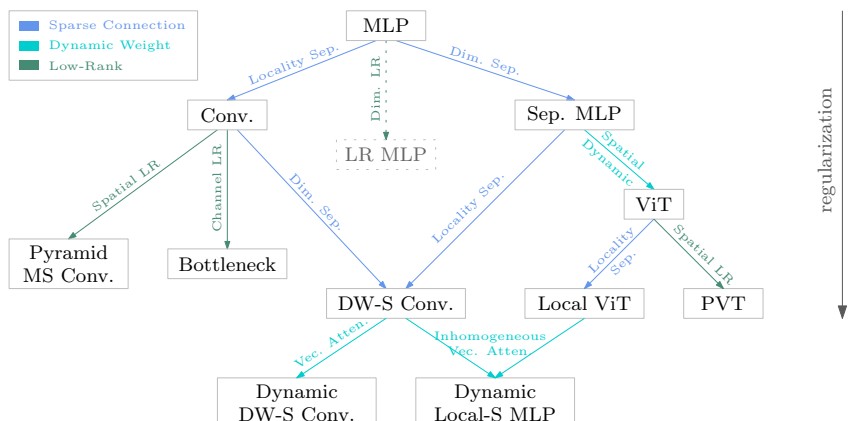

Figure 3: Relation graph for convolution (Conv.), depth-wise separable convolution (DW-S Conv.), Vision Transformer (ViT) building block, local ViT building block, Sep. MLP (e.g., MLP-Mixer and ResMLP), dynamic depth-wise separable convolution (Dynamic DW-S Conv.), as well as dynamic local separable MLP ( e.g., involution (Li et al., 2021) and inhomogeneous dynamic depth-wise convolution) in terms of sparse connectivity and dynamic weight. Dim. = dimension including spatial and channel, Sep. = separable, LR = low rank, MS Conv. = multi-scale convolution, PVT = pyramid vision transformer.

Table 7: The comparison of attention, local MLP (non-dynamic version of local attention, the attention weights are learned as static model parameters), local attention, convolution, depth-wise convolution (DW-Conv.) and the dynamic variant (D-DW-Conv.), as well as MLP and MLP variants in terms of the patterns of sparse connectivity, weight sharing, and dynamic weight. $^{\dagger}$Spatial-mixing MLP (channel-separable MLP) corresponds to token-mixer MLP. $^{\ddagger}1 \times 1$ Conv. is also called point-wise (spatial-separable) MLP. $^{\flat}$The weights might be shared within each group of channels. Please refer to Figure 1 for the connectivity pattern illustration.

| | Sparse between positions | | Sparse between | Weight sharing across | | Dynamic |
| | non-local | full | channels | position | channel | weight |
|---|---|---|---|---|---|---|
| Local MLP | ✓ | | ✓ | | ✓$^{\flat}$ | |
| Local attention | ✓ | | ✓ | | ✓$^{\flat}$ | ✓ |
| DW-Conv. | ✓ | | ✓ | ✓ | | |
| D-DW-Conv. | ✓ | | ✓ | ✓ | | ✓ |
| Conv. | ✓ | | | ✓ | | |
| MLP | | | | | | |
| Attention | | | ✓ | | ✓$^{\flat}$ | ✓ |
| Spatial-mixing MLP$^{\dagger}$ | | | ✓ | | ✓ | |
| $1 \times 1$ Conv.$^{\ddagger}$ | | ✓ | | ✓ | | |

use the window size 3 as an example):

$$\mathbf{W} = \begin{bmatrix} a_2 & a_3 & 0 & 0 & \cdots & 0 & a_1 \\ a_1 & a_2 & a_3 & 0 & \cdots & 0 & 0 \\ \vdots & \vdots & \vdots & \vdots & \ddots & \vdots & \vdots \\ a_3 & 0 & 0 & 0 & \cdots & a_1 & a_2 \end{bmatrix}. \tag{10}$$

For the $C$-channel case, we organize the input into a vector channel by channel: $[\mathbf{x}_1^\top \ \mathbf{x}_2^\top \ \ldots \ \mathbf{x}_C^\top]^\top$, and accordingly the connection weight matrix channel by channel for the $c_o$th output channel, $\mathbf{W}_{c_o} = [\mathbf{W}_{c_o 1} \ \mathbf{W}_{c_o 2} \ \ldots \ \mathbf{W}_{c_o C}]$ (the form of $\mathbf{W}_{c_o i}$ is the same as Equation 10). The whole form could be written as

$$\begin{bmatrix} \mathbf{y}_1 \\ \mathbf{y}_2 \\ \vdots \\ \mathbf{y}_C \end{bmatrix} = \begin{bmatrix} \mathbf{W}_1 \\ \mathbf{W}_2 \\ \vdots \\ \mathbf{W}_C \end{bmatrix} \begin{bmatrix} \mathbf{x}_1 \\ \mathbf{x}_2 \\ \vdots \\ \mathbf{x}_C \end{bmatrix}. \tag{11}$$

**Sep. MLP.** Sep. MLP, e.g., ResMLP and MLP-Mixer, is formed with two kinds of block-sparse matrices: one for channel-mixing and the other for spatial-mixing. In the case that the input is organized channel by channel (the neurons in each channel form a group), $\mathbf{x} = [\mathbf{x}_1^\top \; \mathbf{x}_2^\top \; \ldots \; \mathbf{x}_C^\top]^\top$, the connection weight matrix is in a block-sparse form:

$$\mathbf{W} = \begin{bmatrix} \mathbf{W}_c & \mathbf{0} & \cdots & \mathbf{0} & \mathbf{0} \\ \mathbf{0} & \mathbf{W}_c & \cdots & \mathbf{0} & \mathbf{0} \\ \vdots & \vdots & \ddots & \vdots & \vdots \\ \mathbf{0} & \mathbf{0} & \cdots & \mathbf{0} & \mathbf{W}_c \end{bmatrix}, \tag{12}$$

where the block matrices $\mathbf{W}_c \in \mathbb{R}^{N \times N}$ are shared across all the channels, and the sharing pattern can be modified to share weights within each group of channels.

The input can be reshaped position by position (the neurons at each position forms a group): $\mathbf{x} = [\mathbf{x}_1^\top \; \mathbf{x}_2^\top \; \ldots \; \mathbf{x}_N^\top]^\top$, and similarly one more connection weight matrix can be formulated in a block-sparse form (it is essentially a $1 \times 1$ convolution, $\mathbf{W}_p \in \mathbb{R}^{C \times C}$):

$$\mathbf{W}' = \begin{bmatrix} \mathbf{W}_p & \mathbf{0} & \cdots & \mathbf{0} & \mathbf{0} \\ \mathbf{0} & \mathbf{W}_p & \cdots & \mathbf{0} & \mathbf{0} \\ \vdots & \vdots & \ddots & \vdots & \vdots \\ \mathbf{0} & \mathbf{0} & \cdots & \mathbf{0} & \mathbf{W}_p \end{bmatrix}. \tag{13}$$

The forms of block-sparsity are studied in interleaved group convolutions (Zhang et al., 2017) without sharing the weights across groups.

Sep. MLP can also be regarded as using Kronecker product to approximate the connection matrix,

$$\mathbf{W}\mathbf{x} = \mathrm{vec}(\mathbf{A}\,\mathrm{mat}(\mathbf{x})\mathbf{B}). \tag{14}$$

Here, $\mathbf{W} = \mathbf{B}^\top \otimes \mathbf{A} = \mathbf{W}_c^\top \otimes \mathbf{W}_p$. and $\otimes$ is the Kronecker product operator. $\mathrm{mat}(\mathbf{x})$ reshapes the vector $\mathbf{x}$ in a 2D matrix form, while $\mathrm{vec}(\mathbf{x})$ reshapes the 2D matrix into a vector form. In Sep. MLP, the 2D matrix, $\mathrm{mat}(\mathbf{x}) \in \mathbb{R}^{C \times N}$, is organized so that each row corresponds to one channel and each column corresponds to one spatial position. CCNet (Huang et al., 2019b) and interlaced self-attention (Huang et al., 2019a) use Kronecker product to approximate the spatial connection: the former reshapes the vector in a 2D matrix form along the $x$ and $y$ axes, and the latter reshapes the vector windows by windows.

**Vision Transformer (ViT).** The matrix form is similar to Sep. MLP. The difference is that the matrix $\mathbf{W}_c$ is predicted from each image instance. The weight prediction manner in ViT has a benefit: handle an arbitrary number of input neurons.

**Depth-wise separable convolution.** There are two basic components: depth-wise convolution, and $1 \times 1$ convolution that is the same as channel-mixing MLP in Sep. MLP. Depth-wise convolution can be written in the matrix form:

$$\begin{bmatrix} \mathbf{y}_1 \\ \mathbf{y}_2 \\ \vdots \\ \mathbf{y}_C \end{bmatrix} = \begin{bmatrix} \mathbf{W}_{11} & \mathbf{0} & \cdots & \mathbf{0} \\ \mathbf{0} & \mathbf{W}_{22} & \cdots & \mathbf{0} \\ \vdots & \vdots & \ddots & \vdots \\ \mathbf{0} & \mathbf{0} & \cdots & \mathbf{W}_{CC} \end{bmatrix} \begin{bmatrix} \mathbf{x}_1 \\ \mathbf{x}_2 \\ \vdots \\ \mathbf{x}_C \end{bmatrix}, \tag{15}$$

where the form of $\mathbf{W}_{cc}$ is the same as Equation 10.

**Local ViT.** In the non-overlapping window partition case, local ViT simply repeats ViT over each window separately with the linear projections, applied to keys, values, and queries, shared across windows. In the overlapping case, the form is a little complicated, but the intuition is the same. In the

extreme case, the partition is the same as convolution, and the form is as the following:

$$
\begin{bmatrix} \mathbf{y}_1 \\ \mathbf{y}_2 \\ \vdots \\ \mathbf{y}_C \end{bmatrix} = \begin{bmatrix} \mathbf{W}^d & \mathbf{0} & \cdots & \mathbf{0} \\ \mathbf{0} & \mathbf{W}^d & \cdots & \mathbf{0} \\ \vdots & \vdots & \ddots & \vdots \\ \mathbf{0} & \mathbf{0} & \cdots & \mathbf{W}^d \end{bmatrix} \begin{bmatrix} \mathbf{x}_1 \\ \mathbf{x}_2 \\ \vdots \\ \mathbf{x}_C \end{bmatrix},
\tag{16}
$$

where the dynamic weight matrix $\mathbf{W}^d$ is like the form below:

$$
\mathbf{W}^d = \begin{bmatrix} a_{12} & a_{13} & 0 & 0 & \cdots & 0 & a_{11} \\ a_{21} & a_{22} & a_{23} & 0 & \cdots & 0 & 0 \\ \vdots & \vdots & \vdots & \vdots & \ddots & \vdots & \vdots \\ a_{N3} & 0 & 0 & 0 & \cdots & a_{N1} & a_{N2} \end{bmatrix}.
\tag{17}
$$

**Low-rank MLP.** Low-rank MLP approximates the connection weight matrix $\mathbf{W} \in \mathbb{R}^{D_o \times D_i}$ in Equation 9 using the product of two low-rank matrix:

$$
\mathbf{W} \leftarrow \mathbf{W}_{D_o r} \mathbf{W}_{r D_i},
\tag{18}
$$

where $r$ is a number smaller than $D_i$ and $D_o$

**Pyramid.** The downsampling process in the pyramid networks can be regarded as spatial low rank: $\mathbf{W}(\in \mathbb{R}^{NC \times NC}) \rightarrow \mathbf{W}'(\in \mathbb{R}^{N'C \times N'C})$, where $N'$ is equal to $\frac{N}{4}$ in the case that the resolution is reduced by $\frac{1}{2}$. If the numbers of input and output channels are different, it becomes $\mathbf{W}(\in \mathbb{R}^{NC' \times NC}) \rightarrow \mathbf{W}'(\in \mathbb{R}^{N'C' \times N'C})$.

**Multi-scale parallel convolution.** Multi-scale parallel convolution used in HRNet (Wang et al., 2020; Sun et al., 2019) can also be regarded as spatial low rank. Consider the case with four scales, multi-scale parallel convolution can be formed as as the following,

$$
\mathbf{W} \rightarrow \begin{bmatrix} \mathbf{W}_1 \in \mathbb{R}^{NC_1} \\ \mathbf{W}_2 \in \mathbb{R}^{NC_2} \\ \mathbf{W}_3 \in \mathbb{R}^{NC_3} \\ \mathbf{W}_4 \in \mathbb{R}^{NC_4} \end{bmatrix} \rightarrow \begin{bmatrix} \mathbf{W}'_1 \in \mathbb{R}^{NC_1} \\ \mathbf{W}'_2 \in \mathbb{R}^{\frac{N}{4}C_2} \\ \mathbf{W}'_3 \in \mathbb{R}^{\frac{N}{16}C_3} \\ \mathbf{W}'_4 \in \mathbb{R}^{\frac{N}{64}C_4} \end{bmatrix},
\tag{19}
$$

where $C_1, C_2, C_3,$ and $C_4$ are the numbers of the channels in four resolutions.

## C  Local Attention vs Convolution: Dynamic Weights

We take the 1D case with the window size $2K + 1$ as an example to illustrate the dynamic weight prediction manner. Let $\{\mathbf{x}_{i-K}, \ldots, \mathbf{x}_i, \ldots, \mathbf{x}_{i+k}\}$ correspond to the $(2K + 1)$ positions in the $i$th window, and $\{w_{i-K}, \ldots, w_i, \ldots, w_{i+K}\}$ be the corresponding dynamic weights for updating the representation of the $i$th (center) position. The discussion can be easily extended to multiple weights for each positions, like the $M$-head attention and updating the representations for other positions.

**Inhomogeneous dynamic convolution.** We use the case using only a single linear projection to illustrate inhomogeneous dynamic convolution. The properties we will discuss are similar for more linear projections. The dynamic weights are predicted as the following:

$$
\begin{bmatrix} w_{i-K} \\ \vdots \\ w_i \\ \vdots \\ w_{i+K} \end{bmatrix} = \Theta \mathbf{x}_i = \begin{bmatrix} \boldsymbol{\theta}_{-K}^\top \\ \vdots \\ \boldsymbol{\theta}_0^\top \\ \vdots \\ \boldsymbol{\theta}_K^\top \end{bmatrix} \mathbf{x}_i.
\tag{20}
$$

It can be seen that dynamic convolution learns the weights for each position through the parameters that are different for different positions, e.g., $\theta_k$ corresponds to $w_{i+k}$. It regards the positions in the window as the vector form, keeping the spatial order information.

**Dot-product attention.** The dot-product attention mechanism in the single-head case predicts the weights as the following[6]:

$$
\begin{bmatrix} w_{i-K} \\ \vdots \\ w_i \\ \vdots \\ w_{i+K} \end{bmatrix} = \begin{bmatrix} (\mathbf{x}_{i-K})^\top \\ \vdots \\ (\mathbf{x}_i)^\top \\ \vdots \\ (\mathbf{x}_{i+K})^\top \end{bmatrix} \mathbf{P}_k^\top \mathbf{P}_q \mathbf{x}_i. \tag{21}
$$

Dot-product attention uses the same parameters $\mathbf{P}_k^\top \mathbf{P}_q$ for all the positions. The weight depends on the features at the same position, e.g., $w_{i-k}$ corresponds to $\mathbf{x}_{i-k}$. It in some sense regards the positions in the window as a set form, losing the spatial order information.

We rewrite it as the following

$$
\Theta_d = \begin{bmatrix} (\mathbf{x}_{i-K})^\top \\ \vdots \\ (\mathbf{x}_i)^\top \\ \vdots \\ (\mathbf{x}_{i+K})^\top \end{bmatrix} \mathbf{P}_k^\top \mathbf{P}_q, \tag{22}
$$

from which we can see that the parameters $\Theta_d$ is dynamically predicted. In other words, dot-product attention can be regarded as a two-level dynamic scheme.

Relative position embeddings is equivalent to adding static weights that keeps the spatial order information:

$$
\begin{bmatrix} w_{i-K} \\ \vdots \\ w_i \\ \vdots \\ w_{i+K} \end{bmatrix} = \Theta_d \mathbf{x}_i + \begin{bmatrix} \beta_{-K} \\ \vdots \\ \beta_0 \\ \vdots \\ \beta_K \end{bmatrix}. \tag{23}
$$

A straightforward variant is a combination of the static $\Theta$ and the dynamic $\Theta_d$:

$$
\begin{bmatrix} w_{i-K} \\ \vdots \\ w_i \\ \vdots \\ w_{i+K} \end{bmatrix} = (\Theta_d + \Theta)\mathbf{x}_i. \tag{24}
$$

**Convolutional attention.** We introduce a convolutional attention framework so that it enjoys the benefits of dynamic convolution and dot-product attention: keep the spatial order information and two-level dynamic weight prediction.

---

[6]For presentation clarity, we omit the softmax normalization and the scale in dot-product. What we discuss still holds if softmax and scale are included.

The post-convolutional attention mechanism left-multiplies a matrix (with the kernel size being 3):

$$\Theta_d = \begin{bmatrix} a_2 & a_3 & 0 & 0 & \cdots & 0 & a_1 \\ a_1 & a_2 & a_3 & 0 & \cdots & 0 & 0 \\ \vdots & \vdots & \vdots & \vdots & \ddots & \vdots & \vdots \\ a_3 & 0 & 0 & 0 & \cdots & a_1 & a_2 \end{bmatrix} \begin{bmatrix} (\mathbf{x}_{i-K})^\top \\ \vdots \\ (\mathbf{x}_i)^\top \\ \vdots \\ (\mathbf{x}_{i+K})^\top \end{bmatrix} \mathbf{P}_k^\top \mathbf{P}_q. \tag{25}$$

This can be reviewed as a variant of relative positional embeddings (Equation 23). In the simplified case that the left matrix is diagonal, it can be regarded as the product version of relative positional embeddings (Equation 23 is an addition version).

We can perform a convolution with the kernel size being 3, the kernel weights shared across channels (it is also fine not to share weights), and then do dot-product attention. This is called pre-convolutional attention: perform convolutions on the representations. The two processes are can be written as follows (omit BN and ReLU that follow the convolution),

$$\begin{bmatrix} w_{i-K} \\ \vdots \\ w_i \\ \vdots \\ w_{i+K} \end{bmatrix} = \begin{bmatrix} a_1 & a_2 & a_3 & \cdots & 0 & 0 & 0 \\ 0 & a_1 & a_1 & \cdots & 0 & 0 & 0 \\ \vdots & \vdots & \vdots & \ddots & \vdots & \vdots & \vdots \\ 0 & 0 & 0 & \cdots & a_2 & a_3 & 0 \\ 0 & 0 & 0 & \cdots & a_1 & a_2 & a_3 \end{bmatrix} \begin{bmatrix} (\mathbf{x}_{i-K-1})^\top \\ (\mathbf{x}_{i-K})^\top \\ \vdots \\ (\mathbf{x}_i)^\top \\ \vdots \\ (\mathbf{x}_{i+K})^\top \\ (\mathbf{x}_{i+K+1})^\top \end{bmatrix} \mathbf{P}_k^\top \mathbf{P}_q \begin{bmatrix} \mathbf{x}_{i-1} & \mathbf{x}_i & \mathbf{x}_{i+1} \end{bmatrix} \begin{bmatrix} a_1 \\ a_2 \\ a_3 \end{bmatrix}. \tag{26}$$

It can be generalized to using normal convolution:

$$\begin{bmatrix} w_{i-K} \\ \vdots \\ w_i \\ \vdots \\ w_{i+K} \end{bmatrix} = \mathbf{C}' \begin{bmatrix} \mathbf{x}_{i-K-1} & \mathbf{x}_{i-K-1} & \cdots & \mathbf{x}_{i-K-1} \\ \mathbf{x}_{i-K} & \mathbf{x}_{i-K} & \cdots & \mathbf{x}_{i-K} \\ \vdots & \vdots & \ddots & \vdots \\ \mathbf{x}_i & \mathbf{x}_i & \cdots & \mathbf{x}_i \\ \vdots & \vdots & \ddots & \vdots \\ \mathbf{x}_{i+K} & \mathbf{x}_{i+K} & \cdots & \mathbf{x}_{i+K} \\ \mathbf{x}_{i+K+1} & \mathbf{x}_{i+K+1} & \cdots & \mathbf{x}_{i+K+1} \end{bmatrix} \mathbf{P}_k^\top \mathbf{P}_q \mathbf{C}_3 \begin{bmatrix} \mathbf{x}_{i-1} \\ \mathbf{x}_i \\ \mathbf{x}_{i+1} \end{bmatrix}. \tag{27}$$

Here, **C'** is a $(2K+1)$-row matrix and can be easily derived from the convolutional kernel $\mathbf{C}_3$. The $(2K+1)$ weights, $\{w_{i-1}, w_i, w_{i+1}\}$, correspond to the $(2K+1)$ rows in **C**, respectively. This means that the three positions are differentiated and the same position in each window corresponds to the same row. This explains why the positional embeddings are not necessary when convolutions are adopted (Wu et al., 2021). Using different pairs $(\mathbf{W}_q, \mathbf{W}_k)$ leads to more weights for each position, e.g., $M$ pairs correspond to $M$-head attention.

## D ARCHITECTURE DETAILS

**Overall structures.** Following local vision transformer, Swin Transformer (Liu et al., 2021b), we build two depth-wise convolution-based networks, namely DW-Conv.-T and DW-Conv.-B. The corresponding dynamic versions are D-DW-Conv.-T, D-DW-Conv.-B, I-D-DW-Conv.-T, and I-D-DW-Conv.-B. The depth-wise convolution-based networks follow the overall structure of Swin Transformer. We replace local self attention by depth-wise convolution with the same window size. We use batch normalization (Ioffe & Szegedy, 2015) and ReLU (Nair & Hinton, 2010) instead of layer normalization (Ba et al., 2016) in the convolution blocks.

Table 8: Architectures details of Swin Transformer and depth-wise convolution-based network (DW Conv.) for the tiny model. The architectures for the base model can be easily obtained.

| | downsp. rate (output size) | Swin | | DW Conv. | |
|---|---|---|---|---|---|
| stage 1 | 4× (56×56) | concat 4×4, linear 96-d, LN | | concat 4×4, linear 96-d, LN | |
| | | LN, linear 96x3-d 
 local sa. 7×7, head 3 
 linear 96-d 
 LN, linear 384-d 
 GELU, linear 96-d | × 2 | linear 96-d, BN, ReLU 
 depthwise conv. 7×7, BN, ReLU 
 linear 96-d, BN 
 BN, linear 384-d 
 GELU, linear 96-d | × 2 |
| stage 2 | 8× (28×28) | concat 2×2, linear 192-d , LN | | concat 2×2, linear 192-d , LN | |
| | | LN, linear 192x3-d 
 local sa. 7×7, head 6 
 linear 192-d 
 LN, linear 768-d 
 GELU, linear 192-d | × 2 | linear 192-d, BN, ReLU 
 depthwise conv. 7×7, BN, ReLU 
 linear 192-d, BN 
 BN, linear 768-d 
 GELU, linear 192-d | × 2 |
| stage 3 | 16× (14×14) | concat 2×2, linear 384-d , LN | | concat 2×2, linear 384-d , LN | |
| | | LN, linear 384x3-d 
 local sa. 7×7, head 12 
 linear 384-d 
 LN, linear 1536-d 
 GELU, linear 384-d | × 6 | linear 384-d, BN, ReLU 
 depthwise conv. 7×7, BN, ReLU 
 linear 384-d, BN 
 BN, linear 1536-d 
 GELU, linear 384-d | × 6 |
| stage 4 | 32× (7×7) | concat 2×2, linear 768-d , LN | | concat 2×2, linear 768-d , LN | |
| | | LN, linear 768x3-d 
 local sa. 7×7, head 24 
 linear 768-d 
 LN, linear 3072-d 
 GELU, linear 768-d | × 2 | linear 768-d, BN, ReLU 
 depthwise conv. 7×7, BN, ReLU 
 linear 768-d, BN 
 BN, linear 3072-d 
 GELU, linear 768-d | × 2 |
| stage 4 | 1×1 | LN, AvgPool. 1×1 
 linear classifier | | LN, AvgPool. 1×1 
 linear classifier | |

Table 8 shows the architecture details of Swin Transformer and depth-wise convolution-based networks for the tiny model. Normalizations are performed within the residual block, same as Swin Transformer. The base model is similarly built by following Swin Transformer to change the number of channels and the depth of the third stage.

**Dynamic depth-wise convolution.** Dynamic depth-wise convolution generates the connection weights according to the instance. As described in Section 2.4, for the homogeneous version, we conduct the global average pooling operation to get a vector, and adopt two linear projections: the first one reduces the dimension by $1/4$, followed by BN and ReLU, and then generate the kernel weights and shared for all spatial positions. Unlike SENet (Hu et al., 2018b), we currently do not use the Sigmoid activation function for generating the weights. For the inhomogeneous version, we generate unshared dynamic weight for each spatial position using the corresponding feature. The connection weights are shared across channels to reduce the model parameters and computation complexity. Specifically, we share $3$ and $4$ channels in each group of channels for tiny and base models. Thus the number of model parameters and computation complexity are similar to Swin Transformer.

Table 9: ImageNet classification comparison for ResNet, HRNet, Mixer and ResMLP and gMLP, ViT and DeiT, Swin (Swin Transformer), DW-Conv. (depth-wise convolution), and D-DW-Conv. (dynamic depth-wise convolution). † means that ResNet is built by using two $3 \times 3$ convolutions to form the residual units. Table 7 presents the comparison for representative modules in terms of spare connectivity, weight sharing and dynamic weight.

| method | img. size | #param. | FLOPs | throughput (img. / s) | top-1 acc. | real acc. |
|---|---|---|---|---|---|---|
| *Convolution: local connection* | | | | | | |
| ResNet-38 † (Wang et al., 2020) | $224^2$ | 28M | 3.8G | 2123.7 | 75.4 | - |
| ResNet-72 † (Wang et al., 2020) | $224^2$ | 48M | 7.5G | 623.0 | 76.7 | - |
| ResNet-106 † (Wang et al., 2020) | $224^2$ | 65M | 11.1G | 452.8 | 77.3 | - |
| *Bottleneck: convolution with low rank* | | | | | | |
| ResNet-50 (He et al., 2016) | $224^2$ | 26M | 4.1G | 1128.3 | 76.2 | 82.5 |
| ResNet-101 (He et al., 2016) | $224^2$ | 45M | 7.9G | 652.0 | 77.4 | 83.7 |
| ResNet-152 (He et al., 2016) | $224^2$ | 60M | 11.6G | 456.7 | 78.3 | 84.1 |
| *Pyramid: convolution with pyramid (spatial low rank) features.* | | | | | | |
| HRNet-W18 (Wang et al., 2020) | $224^2$ | 21M | 4.0G | - | 76.8 | - |
| HRNet-W32 (Wang et al., 2020) | $224^2$ | 41M | 8.3G | - | 78.5 | - |
| HRNet-W48 (Wang et al., 2020) | $224^2$ | 78M | 16.1G | - | 79.3 | - |
| *Channel and spatial separable MLP, spatial separable MLP = point-wise $1 \times 1$ convolution* | | | | | | |
| Mixer-B/16 (Tolstikhin et al., 2021) | $224^2$ | 46M | - | - | 76.4 | 82.4 |
| Mixer-L/16 (Tolstikhin et al., 2021) | $224^2$ | 189M | - | - | 71.8 | 77.1 |
| ResMLP-12 (Touvron et al., 2021) | $224^2$ | 15M | 3.0G | - | 76.6 | 83.3 |
| ResMLP-24 (Touvron et al., 2021) | $224^2$ | 30M | 6.0G | - | 79.4 | 85.3 |
| ResMLP-36 (Touvron et al., 2021) | $224^2$ | 45M | 8.9G | - | 79.7 | 85.6 |
| gMLP-Ti (Liu et al., 2021a) | $224^2$ | 6M | 1.4G | - | 72.0 | - |
| gMLP-S (Liu et al., 2021a) | $224^2$ | 20M | 4.5G | - | 79.4 | - |
| gMLP-B (Liu et al., 2021a) | $224^2$ | 73M | 15.8G | - | 81.6 | - |
| *Global attention: dynamic channel separable MLP + spatial separable MLP* | | | | | | |
| ViT-B/16 (Dosovitskiy et al., 2021) | $384^2$ | 86M | 55.4G | 83.4 | 77.9 | 83.6 |
| ViT-L/16 (Dosovitskiy et al., 2021) | $384^2$ | 307M | 190.7G | 26.5 | 76.5 | 82.2 |
| DeiT-S (Touvron et al., 2020) | $224^2$ | 22M | 4.6G | 947.3 | 79.8 | 85.7 |
| DeiT-B (Touvron et al., 2020) | $224^2$ | 86M | 17.5G | 298.2 | 81.8 | 86.7 |
| DeiT-B (Touvron et al., 2020) | $384^2$ | 86M | 55.4G | 82.7 | 83.1 | 87.7 |
| *Pyramid attention: perform attention with spatial low rank* | | | | | | |
| PVT-S (Wang et al., 2021b) | $224^2$ | 25M | 3.8G | - | 79.8 | - |
| PVT-M (Wang et al., 2021b) | $224^2$ | 44M | 6.7G | - | 81.2 | - |
| PVT-L (Wang et al., 2021b) | $224^2$ | 61M | 9.8G | - | 81.7 | - |
| *Local MLP: perform static separable MLP in local small windows* | | | | | | |
| Swin-Local MLP-T | $224^2$ | 26M | 3.8G | 861.0 | 80.3 | 86.1 |
| Swin-Local MLP-B | $224^2$ | 79M | 12.9G | 321.2 | 82.2 | 86.9 |
| *Local attention: perform attention in local small windows* | | | | | | |
| Swin-T (Liu et al., 2021b) | $224^2$ | 28M | 4.5G | 713.5 | 81.3 | 86.6 |
| Swin-B (Liu et al., 2021b) | $224^2$ | 88M | 15.4G | 263.0 | 83.3 | 87.9 |
| *Depth-wise convolution + point-wise $1 \times 1$ convolution* | | | | | | |
| DW-Conv.-T | $224^2$ | 24M | 3.8G | 928.7 | 81.3 | 86.8 |
| DW-Conv.-B | $224^2$ | 74M | 12.9G | 327.6 | 83.2 | 87.9 |
| D-DW-Conv.-T | $224^2$ | 51M | 3.8G | 897.0 | 81.9 | 87.3 |
| D-DW-Conv.-B | $224^2$ | 162M | 13.0G | 322.4 | 83.2 | 87.9 |
| I-D-DW-Conv.-T | $224^2$ | 26M | 4.4G | 685.3 | 81.8 | 87.1 |
| I-D-DW-Conv.-B | $224^2$ | 80M | 14.3G | 244.9 | 83.4 | 88.0 |

# E  SETTING DETAILS

**ImageNet pretraining.** We use the identical training setting with Swin Transformer in ImageNet pretraining for fair comparison. The default input size is $224 \times 224$. The AdamW optimizer (Loshchilov & Hutter, 2019), with the initial learning rate $0.001$ and the weight decay $0.05$, is used for 300 epochs.

The learning rate is scheduled by a cosine decay schema and warm-up with linear schema for the first 20 epochs. We train the model on 8 GPUs with the total batch size 1024. The augmentation and regularization strategies are same as Swin Transformer, which includes RandAugment (Cubuk et al., 2020), Mixup (Zhang et al., 2018a), CutMix (Yun et al., 2019), random erasing (Zhong et al., 2020) and stochastic depth (Huang et al., 2016). The stochastic depth rate is employed as 0.2 and 0.5 for the tiny and base models, respectively, the same as Swin Transformer.

**COCO object detection.** We follow Swin Transformer to adopt Cascade Mask R-CNN (Cai & Vasconcelos, 2019) for comparing backbones. We use the training and test settings from Swin Transformer: multi-scale training - resizing the input such that the shorter side is between 480 and 800 and the longer side is at most 1333; AdamW optimizer with the initial learning rate 0.0001; weight decay - 0.05; batch size - 16; and epochs - 36.

**ADE semantic segmentation.** Following Swin Transformer, we use UPerNet (Xiao et al., 2018) as the segmentation framework. We use the same setting as the Swin Transformer: the AdamW optimizer with initial learning rate 0.00006; weight decay 0.01; linear learning rate decay; 160,000 iterations with warm-up for 1500 iterations; 8 GPUs with mini-batch 2 per GPU. We use the same data augmentation as Swin Transformer based on MMSegmentation (Contributors, 2020). The experimental results are reported as single scale testing.

**Static version of Swin Transformer - Local MLP.** We remove the linear projections applied to keys and queries, accordingly dot production and softmax normalization. The connection weights (corresponding to attention weights in the dynamic version) are set as static model parameters which are learnt during the training and shared for all the images.

**Retraining on** $384 \times 384$**.** We retrain the depth-wise convolution-based network on the ImageNet dataset with $384 \times 384$ input images from the model trained with $224 \times 224$ images. We use learning rate $10^{-5}$, weight decay $10^{-8}$ and stochastic depth ratio 0.1 for 30 epochs for both $7 \times 7$ and $12 \times 12$ windows.

# F  ADDITIONAL EXPERIMENTS AND ANALYSIS

**More results on ImageNet classification.** We give more experimental results with different sparse connection strategies, as shown in Table 9. These results also verify that locality-based sparsity pattern (adopted in depth-wise convolution and local attention) besides sparsity between channels/spatial positions still facilitates the network training for ImageNet-1K.

**Results on large scale pre-training.** Transformers (Liu et al., 2021b; Dosovitskiy et al., 2021) show higher performance compared with the previous convolutional networks with large scale pre-training. We further study the performance on ImageNet-22K pre-training. We first train the model on ImageNet-22K dataset which has about 14.2 million images, and then fine-tune the model on ImageNet-1K classification, downstream detection and segmentation tasks. The same training settings with Swin transformer are used in all tasks. The fine-tuning results in Table 10 and Table 11 indicate the (dynamic) depth-wise convolution based networks could get the performance comparable to Swin transformer with large scale pre-training.

**Cooperating with different normalization functions.** Transformers usually use the layer normalization to stabilize the training, while convolutional architectures adopt batch normalization. We verify different combinations of backbones (Swin and DW Conv.) and normalization functions. The popular used layer normalization (LN), batch normalization (BN), and the dynamic version of batch

Table 10: Comparison on ImageNet-1K classification with ImageNet-22K pre-training.

| | ImageNet-1K fine-tuning | | |
|---|---|---|---|
| | #param. | FLOPs | top-1 acc. |
| Swin-B | 88M | 15.4G | 85.2 |
| DW-Conv.-B | 74M | 12.9G | 84.8 |
| D-DW-Conv.-B | 162M | 13.0G | 85.0 |
| I-D-DW-Conv.-B | 80M | 14.3G | 85.2 |

Table 11: Comparison results on COCO object detection and ADE semantic segmentation with ImageNet-22k pre-training.

| | COCO fine-tuning | | | | | | ADE20K fine-tuning | | |
| --- | --- | --- | --- | --- | --- | --- | --- | --- | --- |
| | #param. | FLOPs | $AP^{box}$ | $AP_{50}^{box}$ | $AP_{75}^{box}$ | $AP^{mask}$ | #param. | FLOPs | mIoU |
| Swin-B | 145M | 986G | 53.4 | 72.1 | 58.1 | 46.1 | 121M | 1192G | 49.4 |
| DW Conv.-B | 132M | 924G | 52.0 | 70.4 | 56.3 | 45.0 | 108M | 1129G | 50.1 |
| D-DW Conv.-B | 219M | 924G | 51.9 | 70.7 | 56.2 | 45.0 | 195M | 1129G | 49.6 |
| I-D-DW Conv.-B | 137M | 948G | 52.9 | 71.2 | 57.2 | 45.8 | 114M | 1153G | 51.3 |

Table 12: Exploring normalization schemes of Swin Transformer and depth-wise convolution based networks (DW Conv.) for the tiny model. The results are reported on the ImageNet top-1 accuracy.

| | Layer Norm. | Batch Norm. | Centering calibrated Batch Norm. | Top-1 Acc. |
| --- | --- | --- | --- | --- |
| Swin | ✓ | | | 81.3 |
| Swin | | ✓ | | 80.9 |
| Swin | | | ✓ | 81.2 |
| DW Conv. | ✓ | | | 81.2 |
| DW Conv. | | ✓ | | 81.3 |
| DW Conv. | | | ✓ | 81.7 |

normalization - centering calibrated batch normalization (Gao et al., 2021) (CC. BN) are verified in the experiments. Table 12 shows the results on ImageNet classification.

**Depth-wise convolution with other architectures.** We conduct experiments on other local attention designs, such as SVT (Chu et al., 2021a) and VOLO (Yuan et al., 2021c) whose implementations are publicly available. SVT uses local self attention as a basic spatial feature fusion operation, while VOLO proposes a new attention module named Vision Outlooker. We replace the local self attention with depth-wise convolution in SVT same as the paper, and replace Vision Outlooker with $7 \times 7$ local self attention and $7 \times 7$ depth-wise convolution, respectively. The remaining structures are unchanged and the same training setting is used as the original papers. The experimental results are shown in Tab 13 and the observations are the same as the Swin Transformer design.

**Retraining on** $384 \times 384$ **images.** Similar to (Liu et al., 2021b), we study the performance of fine-tuning the models: first learn with $224 \times 224$ images, then fine-tune on large images of $384 \times 384$. We study two cases: (1) keep the window size $7 \times 7$ unchanged; and (2) upsample the kernel weights from $7 \times 7$ to $12 \times 12$ as done in (Liu et al., 2021b) for upsampling the relative positional embeddings.

The results are in Table 14[7]. In the case of keeping the window size $7 \times 7$ unchanged, depth-wise convolution (DW) performs better. When using a larger window size $12 \times 12$, depth-wise convolution performs worse than $7 \times 7$. We suspect that this is because upsampling the kernel weights is not a good starting for fine-tuning. In Swin Transformer, using a larger window size improves the performance. We believe that this is because the local attention mechanism is suitable for variable window sizes.

**Cooperating with SE.** Squeeze-and-excitation (Hu et al., 2018b) (SE) is a parameter- and computation-efficient dynamic module, initially designed for improving the ResNet performance. The results in Table 15 show that depth-wise convolution (DW), a static module, benefits from the SE module, while Swin Transformer, already a dynamic module, does not benefit from dynamic module SE. The reason is still unclear, and might lie in the optimization.

## G  POTENTIAL STUDIES

**Complexity balance between point-wise ($1 \times 1$) convolution and depth-wise (spatial) convolution.** Depth-wise convolution takes only about $2\%$ computation in the depth-wise convolution-based architecture. The major computation complexity comes from $1 \times 1$ convolutions. The solutions to this issue could be: group $1 \times 1$ convolution studied in IGC (Zhang et al., 2017; Sun et al., 2018), and

---

[7]Swin Transformer takes slightly higher FLOPs for $7 \times 7$ than $12 \times 12$. The higher computation cost comes from larger padding than $12 \times 12$.

Table 13: Comparison between local attention and depth-wise convolution in VOLO (Yuan et al., 2021c) and SVT (Chu et al., 2021a) architecture. Results are reported on ImageNet classification with tiny model.

| | #param. | FLOPs | top-1 acc. |
|---|---|---|---|
| VOLO-d1 (Yuan et al., 2021c) | 27M | 7.0G | 84.1 |
| VOLO (Local SA)-d1 | 27M | 7.2G | 84.2 |
| DW Conv.-d1 | 26M | 6.9G | 84.2 |
| SVT-S (Chu et al., 2021a) | 24M | 2.8G | 81.7 |
| DW Conv.-S | 22M | 2.7G | 81.9 |

Table 14: Retrain on larger images.

| model | ws. | #param. | FLOPs | Acc. |
|---|---|---|---|---|
| Swin | 7×7 | 28M | 14.4G | 81.8 |
| | 12×12 | 28M | 14.2G | 82.4 |
| DW Conv. | 7×7 | 24M | 11.1G | 82.2 |
| | 12×12 | 25M | 11.5G | 82.1 |

Table 15: Cooperate with SE.

| model | SE | #param. | FLOPs | Acc. |
|---|---|---|---|---|
| Swin | | 28M | 4.5G | 81.3 |
| | ✓ | 29M | 4.5G | 81.2 |
| DW Conv. | | 24M | 3.8G | 81.3 |
| | ✓ | 24M | 3.8G | 81.7 |

channel-wise weighting (like SENet) studied in Lite-HRNet (Yu et al., 2021) and EfficientNet (Tan & Le, 2019; 2021), or simply add more depth-wise (spatial) convolutions.

**Attention weights as channel maps.** Attention weights in attention can be regarded as channel maps. The operations, such as convolution or simple weighting, can be applied to the attention weights. The resT approach (Zhang & Yang, 2021) performs $1 \times 1$ convolutions over the attention weight maps.

**Dynamic weights.** In Swin Transformer and our developed dynamic depth-wise convolution networks, only the spatial part, attention and depth-wise convolution, explores dynamic weights. Lite-HRNet instead studies dynamic weight for point-wise ($1 \times 1$) convolution. It is interesting to explore dynamic weight for both parts.

**Convolution-style MLP weights.** The weights of the spatial-mixing MLP in MLP-Mixer and ResMLP could be modified in the convolution-like style with more weights (some like the relative position embeddings used in local attention, larger than the image window size) so that it could be extended to larger images and downstream tasks with different image sizes.

