# OpenReview forum: "On the Connection between Local Attention and Dynamic Depth-wise Convolution"
_ICLR.cc/2022/Conference — ICLR 2022 Spotlight_

### Official Review · Reviewer_vpCE · 2021-11-01

**Correctness:** 3
**Technical Novelty And Significance:** 3
**Empirical Novelty And Significance:** 3
**Recommendation:** 8
**Confidence:** 4

**Main Review:**

Strengths:
This work is simple and easy to follow. It raises valuable insights on what is the essential components for the "superior" performance of Swin Transformer. With its clear and precise experiments, we can see that the local attention-based networks, Swin Transformer, and the depth-wise convolution-based networks, perform on par in both classification and other downstream tasks such as COCO object detection and ADE semantic segmentation.

I think the single above fact is valuable and enough for acceptance, as it helps us "narrow down what features of other models are most valuable" (comes from a Reddit comment for another work ConvMixer). Besides, this work also analyses three individual properties of local attention and depth-wise Conv, showing more valuable details of their connection.

Weakness:
I am curious about the performance of larger models with the ImageNet 22k dataset, especially for downstream tasks such as detection and segmentation. Based on some current works and my own exps, for the larger size of model and data, attention-based models (Swin, CSwin) do have some advantages on downstream tasks. So I think it would be helpful to include these experiments if the authors have sufficient resources.



**Summary Of The Paper:**

This paper reveals the connection between local attention and dynamic depth-wise convolution. It empirically shows that the models based on depth-wise convolution and the dynamic variants with lower computation complexity perform on-par with or slightly better than Swin Transformer for ImageNet classification and other downstream tasks such as COCO object detection and ADE semantic segmentation.


**Summary Of The Review:**

I think this paper provides valuable insights on the connection between local attention and dynamic depth-wise convolution and I suggest accepting it without a doubt.

---

> ### Author Response · Authors · 2021-11-19
> **Response to Reviewer vpCE**
>
> Q1: *"I am curious about the performance of larger models with the ImageNet 22k dataset, especially for downstream tasks such as detection and segmentation. Based on some current works and my own exps, for the larger size of model and data, attention-based models (Swin, CSwin) do have some advantages on downstream tasks. So I think it would be helpful to include these experiments if the authors have sufficient resources."*
>
> A1: This is a good suggestion. We are collecting resources to conduct the experiments on larger data and models. We afraid it can not be finished during the rebuttal period.
>
> We have some thoughts about that local attention-based models (Swin, CSwin) have advantages on downstream tasks. We think that one reason may include the different window sizes: convolutional networks usually use small kernel size (ResNet, EfficientNet), such as 3x3, while Swin and CSwin use 7x7 windows. As described in [1], the large kernel plays an important role when jointly performing the classification and localization tasks. The recently HRformer [2] also verifies the larger kernel size is important in downstream tasks.
>
>
> [1] Peng, Chao, et al. "Large kernel matters--improve semantic segmentation by global convolutional network." Proceedings of the IEEE conference on computer vision and pattern recognition. 2017.
>
> [2] Yuan, Yuhui, et al. "HRFormer: High-Resolution Transformer for Dense Prediction." Advances in Neural Information Processing Systems. 2021.

---

### Official Review · Reviewer_cAcK · 2021-11-02

**Correctness:** 3
**Technical Novelty And Significance:** 3
**Empirical Novelty And Significance:** 3
**Recommendation:** 8
**Confidence:** 5

**Main Review:**

Strengths

The paper is easy to follow and well-written and builds the connection between local attention and dynamic depth-wise convolution.

Weaknesses

1. Regarding the empirical validation of equivalent, I think simple accuracy is not proper as it summarizes the whole dataset. E.g. prediction disagreement from Swin, mean class accuracy or even centered kernel alignment would be better whether or not the networks act similarly.

2. I am not quite clear about the main difference from Cordonnier et al 2020. The authors pointed out that this work focuses on local attention to depth-wise convolution while the prior art discussed normal convolution and normal self-attention. Nonetheless, the depth-wise convolution is a special case of normal convolution and local attention is a special case of normal attention (local attention is masked attention of normal attention).

3. As depth-wise will slide the window while Swin does not. Why author pick it as an example rather than ViL (Multi-scale vision longformer, Zhang 2021)? ViL adopts both local attention and sliding windows. Or what is the connection to ViL?

**Summary Of The Paper:**

The paper connects the local attention and dynamic depth-wise convolution and validates that empirically.


**Summary Of The Review:**

The paper qualitatively connects local attention and dynamic depth-wise convolution and they validate this connection empirically.

---

> ### Author Response · Authors · 2021-11-19
> **Response to Reviewer cAcK**
>
> We appreciate the constructive review comments and make some clarifications:
>
> Q1: *"Regarding the empirical validation of equivalent, I think simple accuracy is not proper as it summarizes the whole dataset. E.g. prediction disagreement from Swin, mean class accuracy or even centered kernel alignment would be better whether or not the networks act similarly."*
>
> A1: Thanks a lot for your good suggestions. We followed your suggestions and provided the results about the prediction disagreement and mean class accuracy.
>
> First, we report the disagreement ratio of the predictions over the ImageNet validation images. The results are shown in the fourth column of the table below. Second, we report the average absolute difference of class accuracy compared with Swin-T in the fifth column.
>
> For comparison, we also report the disagreement ratio between Swin-T provided by the authors and our reproduced Swin-T. We can observe that the prediction disagreement ratio and the mean class difference between (I-D-)DW Conv. and Swin are similar to those between two reproduced Swin models. For example, the numbers for I-D-DW Conv.-T are 11.5%, and 2.73 , similar to the numbers for Swin-T (reproduced), 10.3% and 2.59.
>
> According to the table, it can be seen that (dynamic) depth-wise convolution and local attention perform equivalently.
>
> |                     | Acc   | Class Acc           | prediction disagreement | Mean Class Difference |
> | ------------------- | ----- | ------------------- | ----------------------- | --------------------- |
> | Swin-T              | 81.16 | 81.16 ($\pm$ 15.25) | 0 %                     | 0%                    |
> | Swin-T (reproduced) | 81.17 | 81.17 ($\pm$ 15.07) | 10.3%                   | 2.59%                 |
> | DW Conv.-T          | 81.27 | 81.27 ($\pm$ 14.86) | 11.6%                   | 2.98%                 |
> | D-DW Conv.-T        | 81.85 | 81.85 ($\pm$ 14.75) | 11.5%                   | 2.71%                 |
> | I-D-DW Conv.-T      | 81.67 | 81.67 ($\pm$ 14.91) | 11.5%                   | 2.73%                 |
>
>
>
> Q2: *"I am not quite clear about the main difference from Cordonnier et al 2020. The authors pointed out that this work focuses on local attention to depth-wise convolution while the prior art discussed normal convolution and normal self-attention. Nonetheless, the depth-wise convolution is a special case of normal convolution and local attention is a special case of normal attention (local attention is masked attention of normal attention)."*
>
> A2: The key difference between Cordonnier et al. and our work lies in the way of building the connections.
>
> In Cordonnier et al., the convolution is simulated by multi-head global self-attention (MHSA) using additional carefully-designed relative positional embedding (RPE), but dropping the attention part.
>
> In our work, we connect (dynamic) depth-wise convolution and local self-attention by connecting the attention weights for self-attention and the dynamic weights for convolution (as well as studying weight sharing). Our way for connection is clearly different from the way in Cordonnier et al. that removes the attention part and uses additional relative positional embeddings.
>
> And the way of transferring normal convolution to depth-wise convolution is to disconnect the channels, while the way of transferring self-attention to local attention is to divide the image into windows where the attention is conducted. This implies that the ways for "special" are also different. Such "specialization" is exactly the sparse connectivity discussed in the paper.
>
> Q3: *"As depth-wise will slide the window while Swin does not. Why author pick it as an example rather than ViL (Multi-scale vision longformer, Zhang 2021)? ViL adopts both local attention and sliding windows. Or what is the connection to ViL?"*
>
> A3: Swin transformer is a purely local self-attention-based framework, almost does not introduce other special components. In their paper, it is also shown that the sliding window strategy performs the same as the shifted window strategy in terms of the performance (we also show it in Table.5 of the original submission). We choose to make comparisons with Swin as the comparison can directly show the difference between local attention and dynamic depth-wise convolution.
>
> In contrast, in ViL, the local self-attention is implemented with the longformer style. This includes both local attention and global attention (global memory) and its interactions. The comparison to ViL might not directly show the difference between local attention and dynamic depth-wise convolution.
>
> The connection to ViL is that the local attention in ViL also resembles dynamic depth-wise convolution. It might be interesting to introduce the global attention into dynamic depth-wise convolution and study the performance.

---

> > ### Comment · Reviewer_cAcK · 2021-11-29
> > **Thanks for the response**
> >
> > The authors addressed my concern and I believe this paper should be accepted. I raise my score to 8.

---

### Official Review · Reviewer_3FL5 · 2021-11-03

**Correctness:** 4
**Technical Novelty And Significance:** 3
**Empirical Novelty And Significance:** 3
**Recommendation:** 8
**Confidence:** 4

**Main Review:**

Well-written and easy-to-follow paper
Forms interesting connections between local attention and dynamic depth-wise convolution in terms of sparse connectivity, weight sharing and dynamic weight computation.
Experimentally demonstrates that dynamic depth-wise convolution is on par or better than local attention in terms of performance on various vision tasks
The proposed inhomogeneous dynamic depth-wise convolution variant is computationally (slightly) better than local attention module

**Summary Of The Paper:**

Recently local attention based vision transformers achieved state-of-the-art results on various visual recognition tasks. This paper rephrases local attention as a channel-wise spatially-locally connected layer with dynamic connection weights. By analyzing local attention form the view of sparse connectivity, weight sharing and dynamic weight computation, the paper discusses the similarities and differences between local attention and dynamic depth-wise convolution. Motivated by this connection, the paper experimentally compares local attention with depth-wise convolution and its dynamic variants on three vision tasks using the macro architecture of Swin-Transformer. The results show that dynamic depth-wise convolution-based models perform on par or better when compared to local attention while being computationally more efficient.


**Summary Of The Review:**

The main difference between local attention and the inhomogeneous dynamic depth-wise convolution (sec 2.4) is in terms of how the dynamic aggregation weights are computed. The on par or superior performance of I-D-DW-Conv when compared to local attention suggests that dot product-based weight computation (which is core to attention)  is not crucial. Dynamic weights that are predicted using only the query/center pixel features also work well. I think this is an interesting for the community to know.

---

> ### Author Response · Authors · 2021-11-20
> **Response to Reviewer 3FL5**
>
> Thanks for your sincere comments. We appreciate your huge efforts for the valuable suggestions. If you have any questions, please let us know.

---

### Decision · Program_Chairs · 2022-01-20

**Decision:**

Accept (Spotlight)

**Comment:**

All three reviewers recommend acceptance. The paper introduces an interesting study and insights on the connection between local attention and dynamic depth-wise convolution, in terms of sparse connectivity, weight sharing, and dynamic weight. The reviews included questions such as the novelty over [Cordonnier et al 2020] and the connection to Multi-scale vision longformer, which were adequately addressed by the authors. The findings in this paper should be interesting to the ICLR community.